# Biomimetic Design for a Dual Concentric Porous Titanium Scaffold with Appropriate Compressive Strength and Cells Affinity

**DOI:** 10.3390/ma13153316

**Published:** 2020-07-25

**Authors:** Han Lee, Jiunn-Der Liao, Yao-Sheng Guo, Yung-Der Juang

**Affiliations:** 1Department of Materials Science and Engineering, National Cheng Kung University, 1 University Road, Tainan 701, Taiwan; rick594007@hotmail.com (H.L.); sunrise10727@gmail.com (Y.-S.G.); 2International Center for Wound Repair and Regeneration, National Cheng Kung University, 1 University Road, Tainan 701, Taiwan; 3Department of Materials Science, National University of Tainan, Tainan 700, Taiwan; juang@mail.nutn.edu.tw

**Keywords:** dual concentric porous, titanium scaffold, bio-mimic design, load-bearable, cell affinity

## Abstract

In repairing or replacing damaged bones, a dual concentric porous titanium scaffold (P-Ti_x-y_) has emerged as a promising bio-mimic design. Herein, various P-Ti_x-y_ were made and sintered with relatively dense (x = 10, 20, or 30% porosity) and loose (y = 45, 55, or 65 porosity) structures. Firstly, NaCl was used as the pore-forming additive and followed by a hydrothermal removal method. The compressive strength of the as-formed P-Ti_x_y_ and surface morphology, nanomechanical property, and cells’ affinity on the cross-sectioned surface of P-Ti_x_y_ (CP-Ti_x_y_) were then characterized. The results demonstrate that the compressive strength of P-Ti_10_45_, P-Ti_20_45_, or P-Ti_20_55_ exhibits a relatively mild decline (e.g., in the range of 181 and 97 MPa, higher than the required value of 70 MPa) and suitable porosities for the intended structure. Nano-hardness on the solid surface of CP-Ti_x_y_ shows roughly consistent with that of CP-Ti (i.e., ~8.78 GPa), thus, the porous structure of CP-Ti_x_y_ remains mostly unaffected by the addition of NaCl and subsequent sintering process. Most of the surfaces of CP-Ti_x_y_ exhibit high fibroblast (L929) cell affinity with low cell mortality. Notably, in the hFOB 1.19 cell adhesion and proliferation test, CP-Ti_20_55_ and CP-Ti_20_65_ reveal high cell viability, most probably relating with the assembly of dual porosities with interconnected pores. Overall, the sample P-Ti_20_55_ provides a relatively load-bearable design with high cell affinity and is thus promising as a three-dimensional bio-scaffold.

## 1. Introduction

Bones come in a variety of sizes and shapes and have a synergetic cortical and spongy structures as organized hard tissues, which have a compressive strength of ~170 MPa, a tensile strength of 104~121 MPa, and a very low shear stress strength of 51.6 MPa [1]. The outside cortical bone with only a few small canals accounts for ~80% of the skeletal mass and acts as a load-bearing support, which may contain relatively stiff trabecular bone. Depending on the species and their types, their Young’s moduli are located in the range of 1~17 GPa [2]. For human cortical bone, it has the ultimate compressive strength ranging from 90 to 200 MPa, while for cancellous bone, it ranges from 0.2 to 10.44 MPa [3]. In spite of relatively low strengths, the spongy bone has honeycomb-like structures with the spaces filling with fluid bone marrow cells, which make blood and some fat cells. In the case of osteoporosis or disease-associated problems, the spongy bone, followed by cortical bone, tends to reduce bone mass that leads to degrade bone quality or structure [4]. Along with aging for example, it is a natural process that bone loss wherein bone breaks down goes faster than bone buildup and less bone remodeling may take place [5,6]. The degradation of bone quality will bring about the decline of overall bone strength. A biomaterial that is competent to re-construct or repair the bone quality is, therefore, demanded. In particular, it will be much advantageous if the added material is likely to mimic a bone structure as a scaffold.

A major choice to compensate the bone mass is to use synthetic porous bio-scaffold to imitate bone structures [7,8,9]. A porous bio-scaffold with osteoconductive properties is capable to promote the migration of bone cells through the secretion of the matrix for bone formation from surrounding bone into the implant site, as a result, it may heal a part of bone structures [10,11]. Currently, there are some metallic and ceramic biomaterials that can be employed in clinical practice [12,13,14,15,16]. The former is suitable as a load-bearing support with tailored surfaces that exhibit minimal reaction with host tissues [17,18]. Together with the required biocompatibility, the metallic devices can be used as implants within bones, joints, and teeth [19,20]. The latter is particularly used as an osteoconductive material, e.g., hydroxyapatite-associated constituents, and designed with varying porosities for firmly ingrowth-bonding with the natural bone [21]. Porous ceramic biomaterials are potentially brittle as compared with metallic ones, therefore, they are usually employed as a coating [18,22,23,24] or a filling material [20] for chemically-induced functions and incorporated with metallic support, which bears, e.g., compression force and fatigue cycles [18,20].

Titanium and titanium-based alloys have appreciated the major interest as one of the most important biocompatible metals. They are in the process of the most common bone stand-in materials and have been intensively used and developed for decades [25,26]. Titanium alloys, e.g., Ti6Al4V usually in a bulk form, have been the most preferred implant materials owing to their durability, stability, resilience, and biocompatibility [26]. However, titanium alloys in powder form to design a porous scaffold are not recommended [27,28]. Pure titanium exhibits not only non-cytotoxic and non-carcinogenic but also biologically inert and excellent resistance to corrosion from a biological perspective. A thin-film titanium dioxide is spontaneously formed after breakage on the titanium surface [29,30]. Some methods can produce porous titanium scaffolds, such as 3D printing to build porous scaffolds, and powder metallurgy-based moulding process. 3D printing is considered as an effective technique that enables direct manufacturing with complex shapes. However, in order to exhibit the biology performance of scaffolds using 3D printing, there is a trick in materials processed.

A porous titanium scaffold is usually produced by titanium foam through a powder metallurgy-based molding process, thereafter, it is formed with relatively low mechanical strength [31,32,33]. In spite, the as-formed scaffold is competent to be used for replacing defective vertebral bodies [33]. In addition, a particularly-designed porous titanium with pores-connectivity is intended for bone and tissue ingrowth into the porous matrix, which has been evaluated by using in vivo histomorphometric analysis [34,35]. For bone defects in complex acetabular revision surgery, a porous titanium is possibly designed to act as a defect filling implant in the case of severe bone loss in the acetabulum [36]. For this application, the porous titanium provides a modulus of elasticity similar to bone, and a coefficient of friction that allows for fitting initial scratch [37]. Therefore, pure titanium with a porous structure has the potential to be custom-made for a purpose in particular for either bone repairing or bone refilling. Also, its high strength/weight ratio makes titanium an alternative material for biodevices that are preferably made from ceramics and polymers, in order to gain higher strength and toughness [38,39,40]. Pure titanium in porous form also displays an excellent property of osseointegration where it connects both structurally and functionally with the underlying bone, however, the major concern is its mechanical strength while it acts as a load-bearing support [18,41].

Three-dimensional construction of a bio-scaffold for tissue repair or replacement has also been developed. For example, a sintered porous titanium with different porosities and pore sizes has been manufactured, in which the amount and size and dimension of the spacer particles, i.e., ammonium hydrogen carbonate, are controlled. [34] A good correlation between these indices and in vivo bone ingrowth has been found [34,42]. The as-formed scaffold has been assessed by considering the effect of narrow pore throats (e.g., fewer than 52 μm [43,44,45]) through two indices that represent the degree of bone and tissue ingrowth into the scaffold. One is the preferable porosity in the range of 48~70% with small pores (e.g., 233–333 μm with a standard deviation (SD) of 105~194 μm [45]). The other is the relation between the porosity or pore size distribution and its load-bearing property. It is known that as the porosity increases, the effective elastic modulus tends to be decreased and thereafter affects the bone ingrowth behavior [46,47]. In addition, the interconnection of pores for bone ingrowth is of great importance. Using hydroxyapatite or β-tricalcium phosphate as a porous scaffold, the interconnection of pores larger than 50 μm is favorable for subsequent mineralized bone formation [48,49]. A porous tantalum welded by a laser is an alternative approach, which is applied for an intervertebral cage product with required condition [38,39,50]. Although an engineered and interconnected pore structure is competent to support bony ingrowth and vascularization, it is most probably oxidized e.g., for pure titanium [51,52]. In spite, to design a three-dimensional bio-scaffold, one can take the intervertebral cage in replacement of a damaged disc as an example. The structure consists of two parts—dense annulus fibrous and loose nucleus pulposus. It is known that stress shielding can be reduced by adjusting the porous structures [53,54,55]. From recent literature, such medical products still focus on solid intervertebral cages or cages with uniform porosity. Intervertebral cages with uniform porosity do not capture the intricate spatial internal microarchitecture of the replaced tissue, thus, a more biomimetic structure is needed.

In brief, a design for a dual porous titanium scaffold with interconnected pores and sufficient load-bearing property is preferable to mimic bone tissue structure. For the former, with the resulting porous matrix, bone cells may take the advantage of interconnected pathways for further proliferation and differentiation. For the latter, the compressive stress of the as-formed porous scaffold should meet the requirement as a bone substitute to support three-dimensional structure. In this study, an innovative design for a three-dimensional dual concentric porous titanium structure is proposed. The custom-made scaffold should fulfill three factors, which could mimic a bone structure, have sufficient load-bearable properties and strong affinity with bone cells. The biocompatibility of the as-designed porous titanium scaffold is then assessed.

## 2. Material and Methods

### 2.1. Preparation of Dual Concentric Porous Titanium

Figure 1a illustrates the formation process of dual concentric porous titanium scaffold P-Ti_x-y_, where the sintered porosities are expected to be x = 10, 20, or 30 (i.e., outer region) and y = 45, 55, or 65% (i.e., inner region). At first, (i) the compressed and pre-sintered titanium and NaCl powders with the weight ratios of x’ = 10, 20, 30 and y’ = 45, 55, 65 were prepared and named as P-Ti_x’_y’_. NaCl (Taiyen Biotech Corp., Tainan, Taiwan) with particle sizes of ~45 μm was selected as the space holder. Titanium (Zhongrui Material Technology Corp., Tainan, Taiwan) and NaCl powders were respectively mixed with 99.8% C_2_H_5_OH (Sigma-Aldrich, St. Louis, MO, USA) through 24-h ball-milling procedure to obtain two as-designed ratios and temporarily separated by a thin-film glutinous (C_6_H_10_O_5_)_n_ (Hsin chemical corp., Tainan, Taiwan) in a mold, followed by a compression force of ~200 MPa for different weight ratios. Note that the powder with lower content of NaCl (x’) is poured into the outer concentric circle area, while that with high content of NaCl (y’) is poured into the inner area; (ii) a heat treatment under vacuum (firstly purged by Argon) at 1000 °C for 3 h, with a heating rate of 4 °C/min and furnace cooling to room temperature, was given to P-Ti_x’_y’_; when sintering the powders, the thin-film glutinous was removed; (iii) to remove the space holder completely, P-Ti_x’_y’_ were put into an autoclave, followed by a hydrothermal process over 100 °C so that water and NaCl could be dissolved under a high-pressure condition; and (iv) the as-formed P-Ti_x_y_, namely, P-Ti_10_45_, P-Ti_20_45_, P-Ti_20_55_, P-Ti_20_65_, and P-Ti_30_65_, were then obtained.

### 2.2. Compression Stress and Porosity of P-Ti_x_y_

Figure 1b(v) illustrates a compression stress test for P-Ti_x_y_ according to the Standard ISO-5833 [56]. A universal electromechanical machine (AG-IS 100 kN, Shimadzu, Japan) was applied with a strain rate of 0.005 mm/mm·min. All tests have been run-up to a strain rate of 50% and a relative compression stress was subsequently determined. A compression test on a sample with the standard size and dimension was implemented. The yield strength and relative strength (i.e., defined as the ratio of the strength of the porous material to that of the solid material) were then obtained.

Bulk porosity of P-Ti_x_y_ was measured using the Archimedes’ method with distilled water impregnation (ASTM C373-88 [57]). Note that A sampling number of 6 (N = 6) were averaged for each mechanically tested. The dimensions S = πr²h, where π = 3.14, r = 5~6 mm, and h = 8~15 mm. Note that it is an overall porosity.

### 2.3. Surface Morphology and Crystalline Structure of the Cross-Sectioned P-Ti_x_y_

Surface morphology of the cross-sectioned dual concentric porous titanium scaffold, denoted as CP-Ti_x_y_, was examined using a field-emission scanning electron microscope (FE-SEM; JSM-7001, JEOL, Tokyo, Japan). The surface of cross-sectioned dual concentric porous titanium scaffold was firstly sputter-coated with Pt and then characterized using an accelerating voltage of 10 kV under vacuum of 5.15 × 10^−3^ Pa. The elemental composition was measured using energy dispersive spectroscope (EDS). The crystalline structure of the surface was also determined using X-ray diffraction (XRD; MiniFlex II, Rigaku, Tokyo, Japan) with CuKα radiation.

### 2.4. Nano-Hardness and Nano-Scratch Tests on the Surface of CP-Ti_x_y_

Nano-hardness of CP-Ti_x_y_ was measured using a nano-indentation system with continuous stiffness measurement (MTS G200, MTS, Palo Alto, CA, USA), which produces highly sensitive load-displacement data at the surface contact level. In the experiment, the triangular pyramid tip of a *Berkovich* diamond indenter with a radius of ~20 nm was accustomed to a controlled relative humidity of 45% at 22 °C. Poisson’s ratio for the surface was set to 0.32. The loading profile was controlled to have a surface approach velocity of 1 nm/s with a sensitivity of 5%. A constant strain rate of 0.05/s at a chosen frequency of 75 Hz was applied. As shown in Figure 1b(vi), a nano-scratch test with a constant loading mode (the head has a load resolution of 1.0 nN) on the surface of CP-Ti_x_y_ was performed. To obtain the quality of seams along the border of the dual porous structures, a traveling distance of ~140 µm crossing over the border was examined.

### 2.5. Cell Affinity on the Surface of CP-Ti_x_y_

Figure 1c(vii) illustrates cell affinity assessment upon (viii) the surface of CP-Ti_x_y_ using fibroblast (L929) or hFOB 1.19 cells. Based on the standard ISO-10993-5 [58], the live/dead L929 cell staining protocol, cell proliferation (MTS) assay, and lactate dehydrogenase (LDH) assay were respectively employed. Earlier toxicological studies used similar cell lines to provide a basis for comparison. The mean cell culture activity provides an assessment of the cells’ overall activity, toxic effects targeting metabolic pathways, and overall viability. L929 cells derived from mouse fibroblast cell line were preserved in alpha modified Eagle’s medium (α-MEM) with 10% horse serum (Gibco, Invitrogen, Carlsbad, CA, USA) and 10 mL of 10^4^ units/mL penicillin-10^4^ μg/mL streptomycin (Sigma, St.0 Louis, MO, USA). Before the experiments, L929 cells were washed with phosphate-buffered saline (PBS) and detached with trypsine (Gibco, Invitrogen). For the MTS assay, the L929 cells were then cultured in a complete medium maintained at 37 °C in a 5% CO_2_ incubator for 24 h, attained to 7.5 × 10^5^ cells/mL in a complete medium, and again maintained at 37 °C under 5% CO_2_ for 24 h. For the MTS and LDH assays, the L929 cells were seeded near confluence (2 × 10^4^ cells/well = 6.75 × 10^5^ cells/mL) on 24-well plates (Nunc, Thermal Scientific, Rochester, NY, USA).

In addition, the human fetal osteoblastic hFOB 1.19 cells line was cultured to do cell viability assay and fluorescence staining on the surface of CP-Ti_x_y_. The manual counting of the hFOB 1.19 cell was performed using the built-in cell-counter plugin of the ImageJ program (NIH, Bethesda, MD, USA). After opening the image to be counted, the cell-counter plugin was opened, internalize selected. The hFOB 1.19 cell were manually counted by the operator through moving the crosshairs over the particle and confirming the identity of the particle by clicking the mouse button.

### 2.6. Statistical Analysis

In this study, all experimental data are normally distributed and are expressed as the mean ± SD. The SD quantifies the amount of variation or dispersion of a set of data around the mean value. When the measurement shows a low SD, it means that the data points tend to be close to the mean of the set; while a high SD happens, it shows the data points are spread out over a wider range of values. Data were analyzed by *Student’s t-*test to establish significance between data points. The values of *p* < 0.05 are considered statistically significant; the smaller the *p*-value, the larger the significance because it means that the hypothesis under consideration may not adequately explain the observation. A sampling number of 6 (N = 6) were averaged for each measurement.

## 3. Results and Discussion

### 3.1. Compression Stress of P-Ti_x_y_

Figure 2a illustrates the measured mixed porosities x + y of P-Ti_x_y_, which also represent the corresponding overall porosities, compared with the theoretical single porosities, x or y, denoted as SP-Ti_x_ or SP-Ti_y_. A rough consistency of resulting porosity by adding NaCl as the space holder was presumably resulted. In Figure 2b, the samples P-Ti_x_y_ with the added NaCl in wt.% (i.e., corresponding to the as-formed porosities) were associated with the measured compression stress and shown as (porosity, strength) in solid line, as compared with SP-Ti_x_ or SP-Ti_y_, shown in dotted line. All the values were respectively averaged (N = 6). The compressive stress of SP-Ti_x_ or SP-Ti_y_ resulted in a sharp decline (i.e., 610 to 162 MPa) with the increase of porosity. For single porosity lower than e.g., ~26%, a compressive stress lower than 70 MPa could be resulted. According to ISO 5833, for an implantable and load-bearing scaffold, its compression stress should be higher than 70 MPa [59,60]. For P-Ti_x_y_, a combined porosity lower than e.g., ~38%, a compressive stress higher than 70 MPa was still remained; a relatively mild decline from 592 to 97 MPa was resulted. For the cases of P-Ti_10_45_, P-Ti_20_45_, and P-Ti_20_55_, their compression stresses with respect to averaged porosities were thus relatively suitable for the intended structure of the dual-concentric porous design.

### 3.2. Surface Morphology and Crystalline Structure of CP-Ti_x_y_

In Figure 3a–f, SEM photo-graphs of various surfaces of CP-Ti_x_y_ was shown. The indicated boundaries for the dual porous structures (i.e., x and y) were marked. Note that inside the dotted lines is the region y with a relative high porosity. SEM photo-graphs from (b) to (f) showed their pore-size morphologies, e.g., the inner region y ranged from ~45 to ~65%. The corresponding EDS spectra within the regions y, marked with red squares (#2~6 with the reference #1) were shown in Appendix A. No significant change of the element Ti was found. By estimation, the pore sizes of dual porous structures ranged from ~80 to ~650 μm, as shown in Appendix A while the surface roughness was from ~20 to ~90 nm, as shown in Appendix A. Note that the connectivity of pores on the surface of CP-Ti_x_y_ is anticipated to enhance affinity with cells [34,41].

The crystalline structures of CP_Ti and CP-Ti_x_y_ were examined. In Figure 4a, the peaks at 2θ = 35° and ~63.5°, which appear at 1000°C, are respectively assigned to the (100), (002), (010), (101), (102), and (110). The diffraction peaks are the reflections of tetragonal Ti (JCPDS card No. 44-1294) and the same for all samples. In Figure 4b, lattice constants a and c for Ti structure of CP-Ti_x_y_ were confirmed without significant difference. It reveals that the addition of NaCl particles did not cause any alteration in the lattice parameters of Ti structure.

### 3.3. Nanomechanical Property of CP-Ti_x_y_

In Figure 5a, nano-hardness of CP-Ti_x_y_ was shown. The measured values for CP-Ti, CP-Ti_10_45_, CP-Ti_20_45_, CP-Ti_20_55_, CP-Ti_20_65_, and CP-Ti_30_65_ were 8.78 ± 0.35, 9.04 ± 0.83, 8.60 ± 0.19, 9.42 ± 0.8, 8.49 ± 0.82, and 7.81 ± 1.00 GPa, respectively. Note that the indented sites are usually located upon a solid part of CP-Ti_x_y_. As compared with the surface of CP-Ti (i.e., 8.78 ± 0.35 GPa, the dotted a), nano-hardness on the solid surfaces of CP-Ti_x_y_ remain affected, except for CP-Ti_30_65_, by the addition of NaCl and subsequent sintering process. A reference nano-hardness value of 0.52 ± 0.15 GPa for osteons and 0.59 ± 0.20 GPa for interstitial bone tissue were marked in the dotted lines b and c [61]. The effect of stress shielding for P-Ti_x_y_ in bone tissue is expected to be much relaxed, as compared with that for a metallic biomaterial, e.g., Ti6Al4V [62,63]. In Figure 5b, nano-scratch tests along the border between the regions x (start point, S) and y (finish point, F) were performed and shown. Under a constant-load mode of nano-scratch, no significant change by verifying the penetration depths, along the border of regions x and y, was found. Thus, it indicates that a continuity of dual porous structures is made, which may thus contribute to the overall compressive strength of P-Ti_x_y_, as measured in Figure 2b.

### 3.4. Cells’ Affinity upon the Surface of CP-Ti_x_y_

As shown in Figure 2b, Figure 3, and Figure 5a, the samples CP-Ti_20_45_, CP-Ti_20_55_, and CP-Ti_20_65_ were all suitable for current design of dual porous structure. Further cell affinity tests on the above surfaces were assessed. Based on the quantitative assessment of ISO 10993-5, cytotoxicity can be classified through MTS and LDH tests and the degree of morphologies of fibroblasts. In Figure 6, by taking the surface of CP-Ti as the control group, significant differences (*p* < 0.05) were resulted for all P-Ti_x_y_ samples, except for MTS assay on the surface of CP-Ti_30_65_. In most cases, cell viability on the surfaces of CP-Ti_x_y_ is significantly enhanced, as compared with that of CP-Ti. As also agreed with the tests of compressive stress of P-Ti_x_y_ and nano-hardness and nano-scratch of CP-Ti_x_y_, the samples CP-Ti_20_45_, CP-Ti_20_55_, and CP-Ti_20_65_ thus remain suitable for current design. On the other hand, at an early stage test, the levels of LDH leakage for the surfaces of CP-Ti_x_y_ demonstrated significantly reduced as compared with that of CP-Ti. Note that the LDH assay is a colorimetric detection of cellular toxicity through lactate dehydrogenase release into the culture medium [64]. The testing cells are presumably affected by the composition of the contact surface, which may lead to lipid peroxidation and sub-lethal effects on the membranes of the cells and the effects of LDH leakage are a result of the formation of pores in the cell membrane. The result also implies that the cell toxicity on the surface of CP-Ti_x_y_ is presumably less than that of non-porous CP-Ti. In this study, titanium can form a superficial oxide layer that osteoblast attachment and osteointegration. In MTS assay, the cell viability increased as the porosity of porous titanium increased. In L929 cell viability assay, the relationship between porosity and cell viability showed a similar trend with MTS assay, but the cell viability started to decrease as the porosity went too high (70%). When the porosity goes too high, it may cause a decrease in cell viability.

Cell contact, attachment, and subsequent adhesion upon the surface of CP-Ti_x_y_ are among the earlier phases of cell-material interactions [65,66] that profoundly influence integration with a hard tissue and eventually lead to success or failure of a bio-scaffold [67]. For these purposes, cell-adhesion and -proliferation behaviors can be related with the measurement of attached cell numbers, while a so-called “time-dependent phenotypic response” of a model osteoblast cell line—hFOB 1.19 to CP-Ti_x_y_ is particularly assessed [67,68]. In this study, cell-attachment and proliferation behaviors are associated using morphology as an indicator of cell phenotype. The hFOB 1.19 behaviors in contact with the surfaces of CP-Ti_x_y_ is to have the obvious benefit of understanding cell adhesion on osteoblast interactions with the surfaces. In Figure 7a, the fluorescence images after 3 days incubation via the hFOB 1.19 cell viability assay were shown. The hFOB 1.19 cell numbers on the surface of region y (i.e., higher porosity area) revealed at least a two-fold increase, as compared with that of region x (i.e., lower porosity area). In Figure 7b, significant increases (at least two folds) of averaged cell numbers (i.e., from regions x and y) has been found from the overall surfaces of CP-Ti_20_55_ and CP-Ti_20_65_. It is clearly shown that a dual porous surface contributes to cell adhesion and proliferation, whereas an appropriate control on the pore size is also vital owing to incorporation of pores-connectivity in the two regions. In comparison with MTS and LDH tests, as shown in Figure 6, the hFOB 1.19 cell test to the surfaces of CP-Ti_x_y_ results in a relatively sensitive consequence. Moreover, in view of previous assessments such as bulk strength, shown in Figure 2b and nano-hardness on the solid surface of CP-Ti_x_y_, shown in Figure 5a, the sample CP-Ti_20_55_ exhibits the most preferred choice for our current design.

## 4. Conclusions

In this work, dual concentric porous titanium scaffolds (P-Ti_x-y_) with various porosities have been designed, fabricated, and evaluated. Firstly, according to the compression test and the ISO 5833, the porosities with respect to their compressive strengths have accessed the standard for an implantable and load-bearing scaffold. Secondly, a continuous border between two porosities is confirmed by a nano-scratch test on the surfaces of the cross-sectioned P-Ti_x-y_(CP-Ti_x-y_). Thirdly, the surfaces of CP-Ti_x-y_ have been investigated by morphological and nanomechanical tests, followed by cell affinity assays. For the former, the pores distribution and the compressive strength of solid titanium structure are suitable as a durable porous scaffold. For the latter, the standard ISO-10993-5, the live/dead L929 cell staining protocol, cell proliferation (MTS) assay, and lactate dehydrogenase (LDH) assay have been respectively employed. In addition, a model osteoblast cell line—hFOB 1.19 to CP-Ti_x_y_ is particularly assessed. Considering the connectivity of pores in the higher and lower porosity-regions, appropriate integration of pore sizes in the dual porous structure also plays an important role. Overall, P-Ti_20_55_ provides a promising load-bearable property with high cell affinity and thus has great potential for designing a biomimetic scaffold.

## Figures and Tables

**Figure 1 materials-13-03316-f001:**
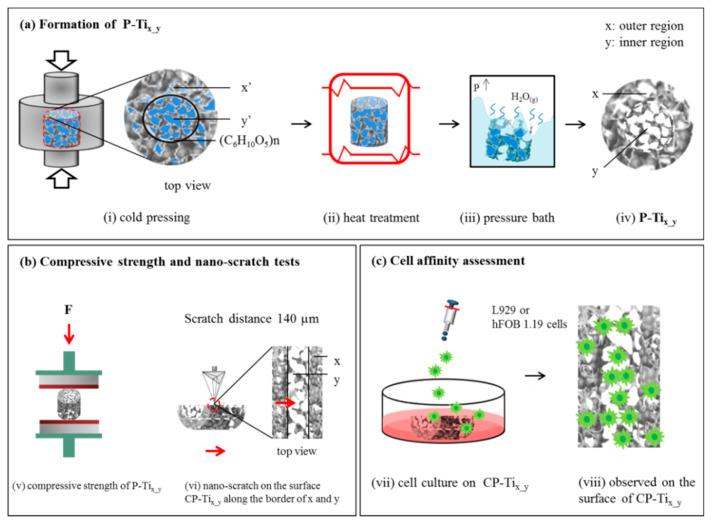
(**a**) An illustration of forming dual porous titanium scaffold (P-Ti_x_y_): (**i**) Cold pressing to form the pre-sintered, NaCl-added samples—P-Tix’_y’—where pre-sintered (outer and inner) porosities, x’ = 10, 20, or 30% and y’ = 45%, 55%, or 65%. Note that a very thin film (C_6_H_10_O_5_)_n_ was used to temporarily separate the dual porous titanium powders; (ii) a heat treatment at 1000 °C for 2 h under vacuum to sinter titanium powders and removing the thin film; (iii) a pressure bath to remove NaCl additive; (iv) then the formation of P-Ti_x_y_. (**b**) An illustration to evaluate the samples P-Ti_x_y_: (v) a compression stress test, and (vi) a nano-scratch test on the surface of the cross-sectioned P-Ti_x_y_ (CP-Ti_x_y_) with a traveling distance of 140 µm along the border of dual porous structures. (**c**) An illustration of cell affinity assessment upon the surface of CP-Ti_x_y_, using L929 or hFOB 1.19 cells.

**Figure 2 materials-13-03316-f002:**
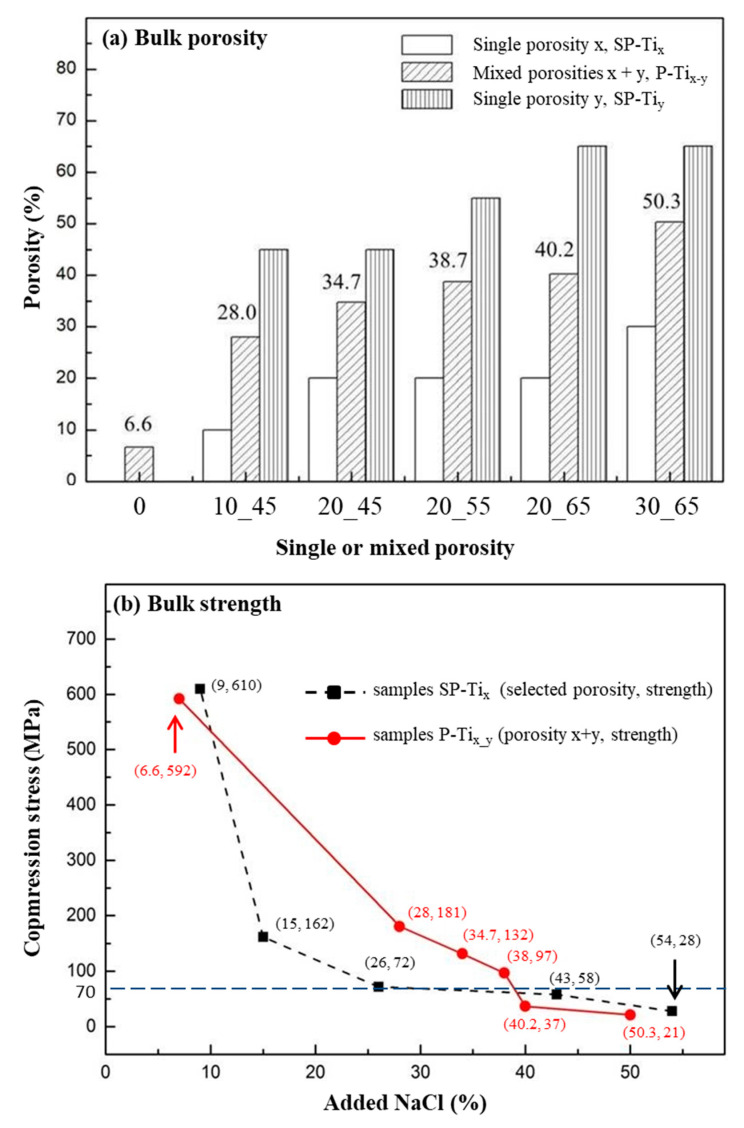
(**a**) The experimental values of mixed porosities, x + y of P-Ti_x_y_, compared with the theoretical values of single porosity, x or y (SP-Ti_x_ or SP-Ti_y_). (**b**) The measured compression stress of P-Ti_x_y_ (in solid line) was compared with that of P-Ti of different porosities (in dotted line). The individual values were obtained and expressed as (porosity, strength).

**Figure 3 materials-13-03316-f003:**
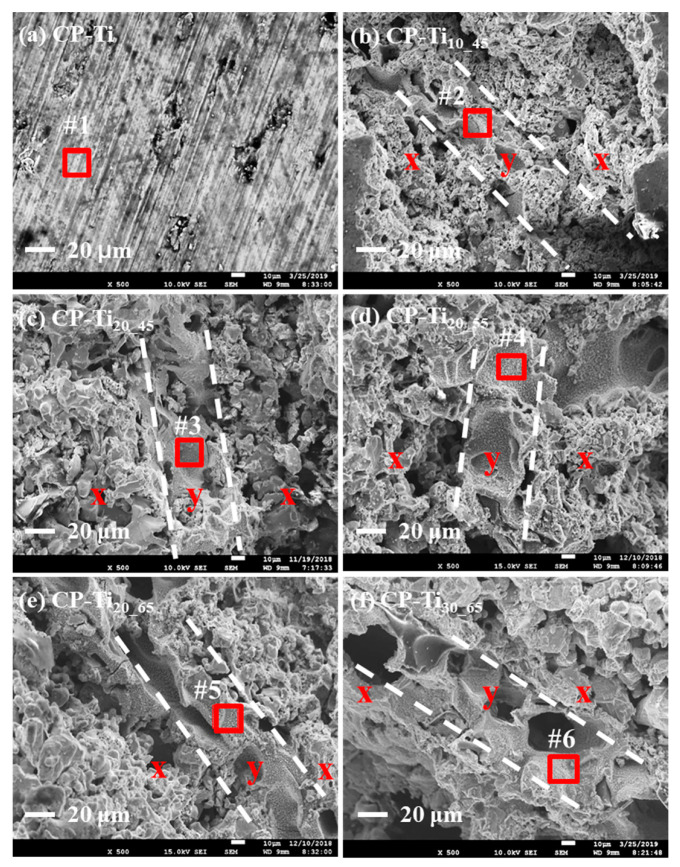
Scanning electron microscope (SEM) photo-images of dual porous structures: (**a**) CP-Ti, (**b**) CP-Ti_10_45_, (**c**) CP-Ti_20_45_, (**d**) CP-Ti_20_55_ (**e**) CP-Ti_20_65_, and (**f**) CP-Ti_30_65_. The morphologies (**b**)~(**f**) indicate the boundaries (i.e., denoted as the two dotted lines) of the dual porous structures (denoted as x and y) from the surfaces of P-Ti_x_y_. The corresponding EDS spectra within the region y, respectively marked with red squares (#1~6), were shown in Appendix A.

**Figure 4 materials-13-03316-f004:**
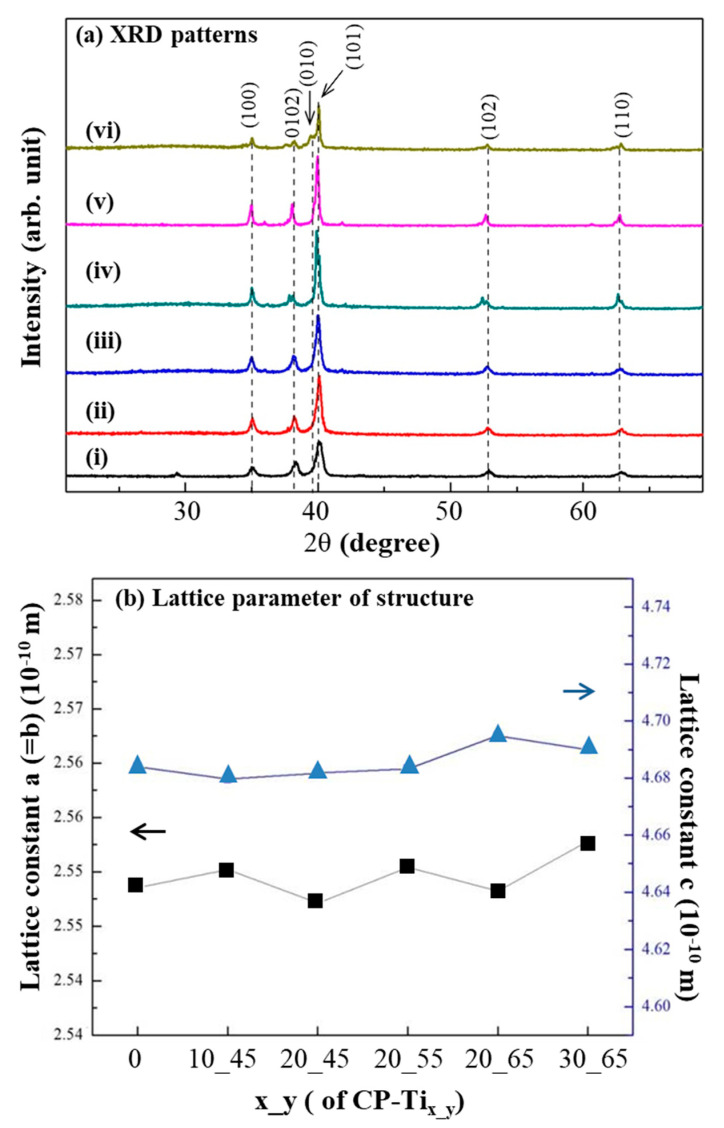
(**a**) X-ray diffraction (XRD) patterns from the surfaces of: (i) CP-Ti, (ii) CP-Ti_10_45_, (iii) CP-Ti_20_45_, (iv) CP-Ti_20_55_, (v) CP-Ti_20_65_, and (vi) CP-Ti_30_65_. No obvious difference of their XRD pattern was resulted. (**b**) Lattice constants a and c from the above samples were also calculated without significant difference.

**Figure 5 materials-13-03316-f005:**
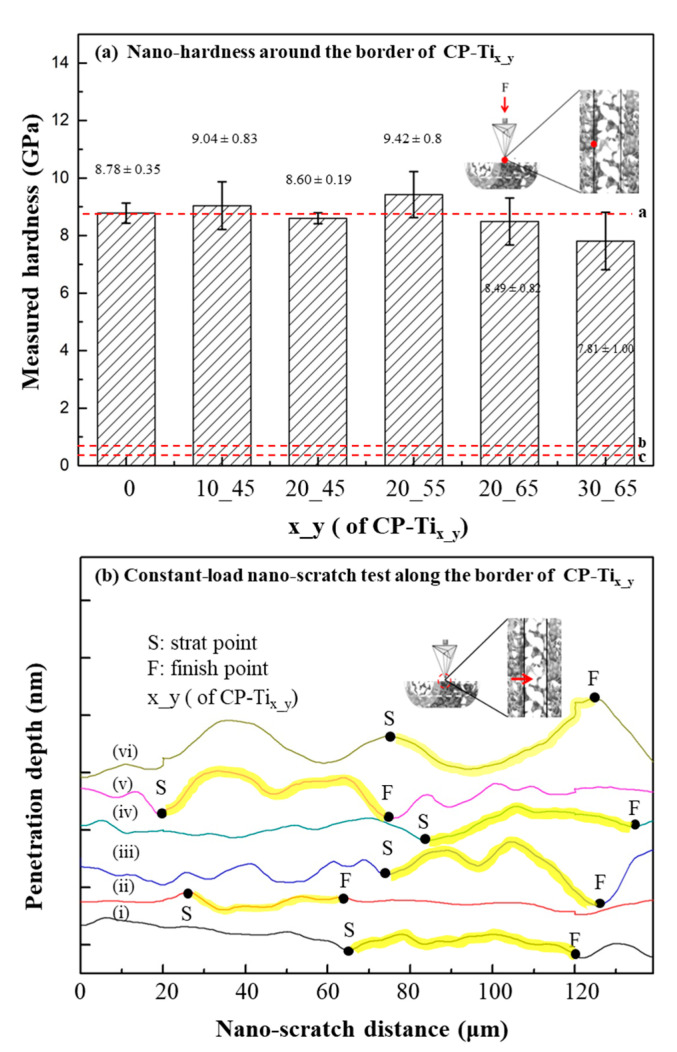
(**a**) Nano-hardness tests around the border (i.e., one of the dotted lines as shown in Figure 2) of dual porous structures. (**b**) Constant-load nano-scratch test along the border (i.e., crossing over the dotted line with the direction from regions x to y). The scratch distances were marked from the starting point S and the final point F. In general, under a constant load mode, no significant change in the penetration depth was found. As a result, a continuous dual porous structure was obtained.

**Figure 6 materials-13-03316-f006:**
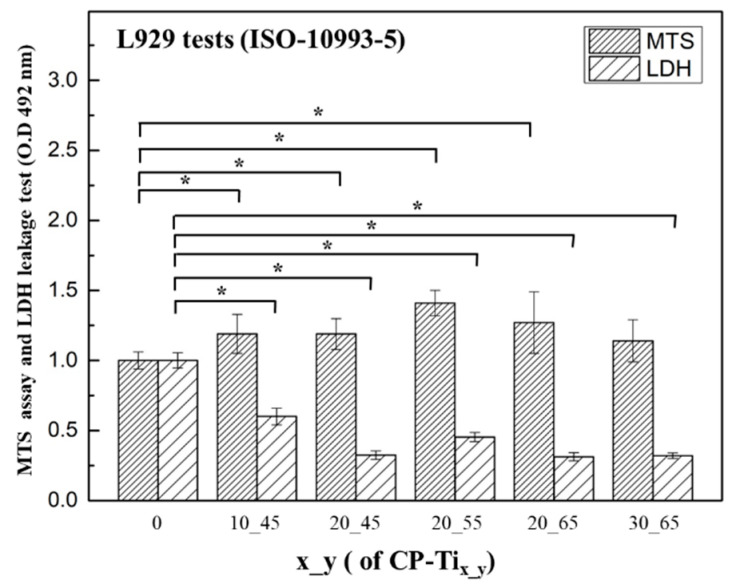
L929 Cell affinity was examined by MTS assay and lactate dehydrogenase (LDH) leakage test upon the surfaces of CP-Ti_x_y_: CP-Ti_10_45_, CP-Ti_20_45_, CP-Ti_20_55_, and CP-Ti_20_65_. For MTS assays, significant increases on cell viability were found on all the surfaces. For LDH leakage tests, significant reduction on cell death was found, except for the surface of CP-Ti_10_45_. *: *p* < 0.05 had statistical differences.

**Figure 7 materials-13-03316-f007:**
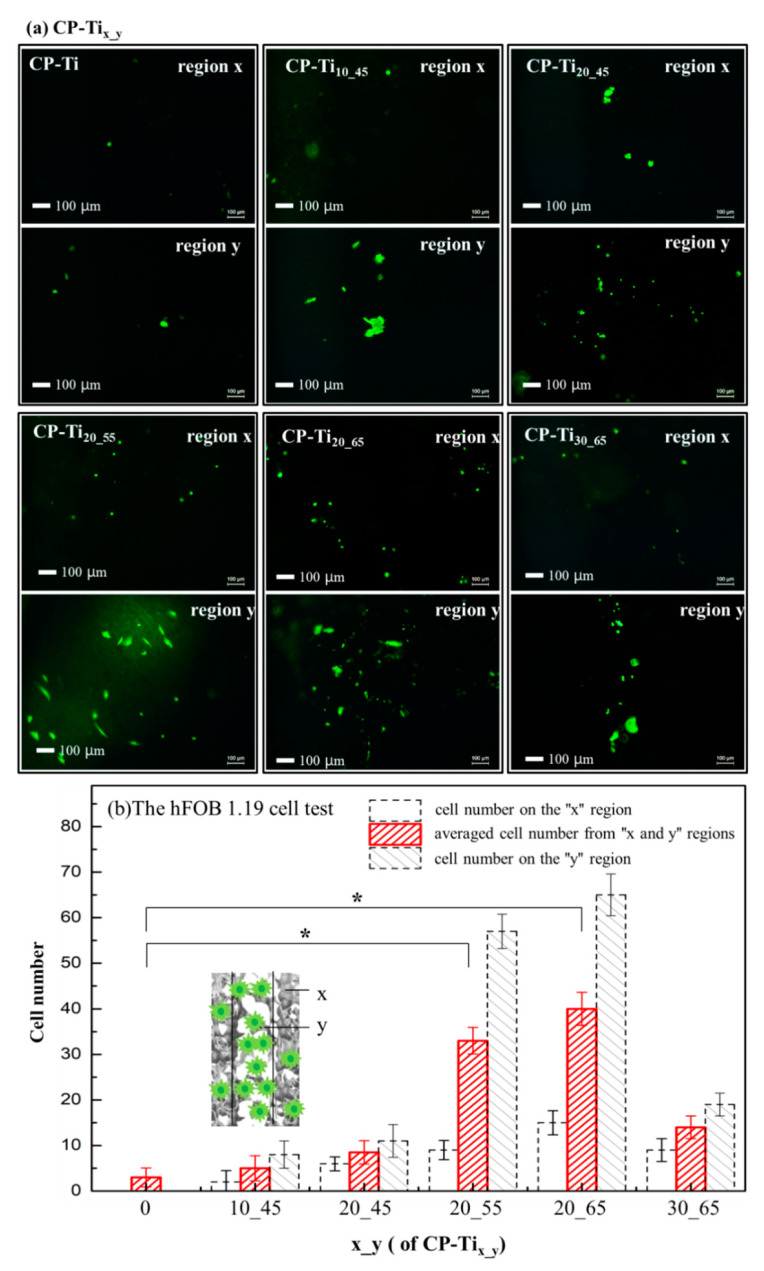
(**a**) The hFOB 1.19 cell viability assays, examined by taking fluorescence images after 3 days incubation upon the surfaces of CP-Ti_x_y_: CP-Ti, CP-Ti_10_45_, CP-Ti_20_45_, CP-Ti_20_55_, CP-Ti_20_65_, and CP-Ti_30_65_. Cell numbers were counted on the regions of x (low porosity), y (high porosity), and around the border (i.e., the dotted lines shown in Figure 2). (**b**) By averaging the cell numbers from the regions x and y, the relation between cell numbers and the overall surfaces were demonstrated. It shows that cell viability on the surfaces of CP-Ti_20_55_ and CP-Ti_20_65_ significantly increases. *: *p* < 0.05 had statistical differences.

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
