# Peer review of "Biomimetic Design for a Dual Concentric Porous Titanium Scaffold with Appropriate Compressive Strength and Cells Affinity"

_materials, 2020, doi:10.3390/ma13153316_

Round 1
Reviewer 1 Report
Review of the article materials-869885, Title: Biomimetic design for a three-dimensional dual concentric porous titanium scaffold with appropriate compressive strength and cells affinity
Authors: Han Lee, Jiunn-Der Liao, Yao-Sheng Guo, Yung-Der Juang
The paper is interesting and the tackled topic falls within the scope of the Journal. The Authors fabricated different samples of dual concentric porous titanium scaffolds and characterized them from the mechanical and the biological point of view. They found that a specific scaffold type provides a relatively load-bearable design with high cell affinity and is thus promising as a three-dimensional bio-scaffold. The paper, in my humble opinion deserves publication should the Authors be prepared to address the following comments.
Line 19. What do the Authors mean by compression stress? Perhaps they meant compression strength otherwise, they should clarify the value of load for which such a stress was measured.
Line 21. "Nano-hardness on the solid surface of CP-Tix_y shows roughly consistent with that of CP-Ti (i.e., ~8.78 GPa), Thus, the porous structure..." should read "Nano-hardness on the solid surface of CP-Tix_y shows roughly consistent with that of CP-Ti (i.e., ~8.78 GPa), thus, the porous structure..."
Line 37. “For human femur bone along its length, the inner spongy bone has the ultimate compressive and tensile strengths of ~205 and ~135 MPa, respectively. [3]”. Are the Authors sure of this sentence? From the paper [3] they mention I found that these properties refer to the cortical bone.
Line 39. The reference [3] should be moved before the point. The same correction should be implemented throughout the entire manuscript.
Line 107. In my opinion a more exhaustive review of the state of the art regarding the design and the optimization of functionally graded scaffolds should be carried out by the Authors.
Figure 1. How did the authors measure the strength? Did they refer to the highest point of the stress-strain curve?
Line 147. Please, clarify the number and the dimensions of the samples mechanically tested
Line 191. “The SD can magnanimity data and dispersion of a set or quantify the amount of variation of data.” It is a little bit difficult to follow this sentence. Please, rephrase.
Figure 2. I am not clear which are the values of porosity for which the reported strength values were measured.
Line 246. The use of acronyms throughout the manuscript makes the reading of the text difficult. Please, minimize the use of acronyms as much as possible.
Line 255. A possible value of nano-hardness for this Ti alloy should be indicated in the paper.
Figure 5(b). The yellow part is not correctly indicated for the curve (ii)
Author Response
Dear Editor,
We highly appreciate the reviewers’ comments that help to improve the quality of the revised manuscript. Their addressed questions have been carefully replied.
Attached please find 4 files: the replies to reviewers, the list of changes, supporting data, and the secondly revised manuscript.
Thank you very much for your kind consideration to publish this paper in a regular issue.
Yours sincerely,
Jiunn-Der Liao
The corresponding author

Reviewer 2 Report
Dear authors, I read your research. it is well conducted. Nevertheless, I just suggest some minor changes to make this study more readable for all the readers.
Abstract should be structured. Study design should be easily recognized (maybe also in the title). Aim, materials and methods and results easily readable in the abstract.
Introduction section should be shortened.
Materials and methods should be improved adding some informations about the locations, relevant dates, study design...
exposure, follow-up, and data collection
Conclusion should be more coincise limiting to the conclusion from the present study avoiding repetition of materials and methods as well as results.
English language need improvements
Author Response

(The authors gave the same response as above.)

Reviewer 3 Report
The manuscript topic is actual and the paper has merit. It could be attractive, adequate and interesting for the journal readers. However there are some point that authors should address in order to have a final more complete paper. Authors should underline the limitation of the value of the study, and the clinical implication of the presented study should be added. At this stage the paper seems to be directed to scientist and not clinicians. Please emphasize the clinical application of the study, and its scientific rationale.
The final aim of the paper is not well focused as well as the possibile application of the titanium scaffold. Firstly and Secondary outcomes should be better highlighted in order to have a more complete and attractive paper.
Pleas increase introductions and discussion section.
Author Response

(The authors gave the same response as above.)

Reviewer 4 Report
Dear Authors,
In my opinion the manuscript titled "Biomimetic design for a three-dimensional dual concentric porous titanium scaffold with appropriate compressive strength and cells affinity" is interesting and in general well written, however before acceptance needs minor revision.
Abstract: this section is well written.
Introduction: this section is clear and well written. The citation of the references is comprehensive.
"2. Experimental section" - I suggest to use the subtitle as: "2. Material and methods"
Results and discussion: this section is well described.
All the figures are clear and in good quality.
The supplementary material includes the support of the results.
Please check the manuscript again carefully- I found several misspellings.
Author Response

(The authors gave the same response as above.)

Round 2
Reviewer 1 Report
Authors have fully addressed my concerns. The paper now can be published in Materials.
Reviewer 3 Report
Authors made excellent job addressing all the reviewer's note and requests